



# Using automatic calibration to improve the physics behind complex numerical models: An example from a 3D lake model using Delft3d (v6.02.10) and DYNO-PODS (v1.0)

Marina Amadori[1,2], Abolfazl Irani Rahaghi[3,4], Damien Bouffard[3,5], and Marco Toffolon[2]

[1]Institute for Electromagnetic Sensing of the Environment, National Research Council, Milan, Italy
[2]Department of Civil, Environmental and Mechanical Engineering, University of Trento, Trento, Italy
[3]Eawag, Swiss Federal Institute of Aquatic Science and Technology, Surface Waters – Research and Management, Switzerland
[4]Department of Geography, University of Zurich, 8057, Zurich, Switzerland
[5]Faculty of Geosciences and Environment, Institute of Earth Surface Dynamics, University of Lausanne, Geopolis, Mouline, 1015, Lausanne, Switzerland
**Correspondence:** Marina Amadori (marina.amadori@unitn.it)

**Abstract.** Models are simplified descriptions of reality and are intrinsically limited by the assumptions that have been introduced in their formulation. With the development of automatic calibration toolboxes, finding optimal parameters that suit the environmental system has become more convenient. Here, we explore how optimization toolboxes can be applied innovatively to uncover flaws in the physical formulations of models. We illustrate this approach by evaluating the effect of simplifications
embedded in the formulation of a widely used hydro-thermodynamic model. We calibrate a Delft3D model based on temperature profiles for a case study, Lake Morat (Switzerland), through the optimization tool DYNO-PODS. Results show that neglecting the fraction $\beta$ of shortwave radiation absorbed at the water surface can be compensated by higher values of the light extinction coefficient. This leads to unrealistic values of the latter parameter, as the optimization pushes the coefficient towards the limit of no transparency, consistent with the need to reproduce a significant absorption at the surface. While it is
well-known that $\beta$ is significantly larger than zero, its absence in the model was never noticed as critical. The extensive use of automatic calibration tools may offer similar outcomes in other applications.

## 1 Introduction

Numerical models serve as powerful tools capable of embracing the complexity of intricate environmental dynamics. In many branches of environmental science, such as meteorology, climatology, hydrology, oceanography and limnology, thermo-
hydrodynamic models have become standard tools to simulate and understand specific physical processes. These models are rooted on the numerical solution of physical first principles, e.g. the mass, momentum, and heat equations. Although first principles are well established, modelling the physics of fluid systems remains complicated for two reasons. First, the grid size limits the range of applicability of models based on first principles (e.g., Direct Numerical Simulations, DNS) and poses the challenge to properly parameterize associated sub-grid scale processes. This implies that models that were developed for





a specific environment, where the parameterization was adequate, may not be optimized for other contexts characterized by
different spatial scales. Second, imposing the boundary conditions is a complicated task as the forcing acting at the boundaries
of the computational domain is often only partially known. For instance, the estimation of the surface heat fluxes is based on
parameterizations that have to cope with both the complex physics and the uncertainties associated to the forcing. Contrary
to the first principles solved by the model, such boundary conditions often lack universality. In the specific case of models

widely adopted to simulate three-dimensional lake dynamics, many were originally developed for marine, riverine or estuarine
environments. This is the case of, e.g., Delft3D (Lesser et al., 2004), MITgcm (Marshall et al., 1997), ROMS (Shchepetkin and
McWilliams, 2005), FVCOM (Chen et al., 2003), NEMO (Madec et al., 2023), POM (Blumberg and Mellor, 1987), among
a few others. Therefore, some processes that are crucial in lakes, but not for the original environment, may not be correctly
reproduced. Scientists thus must critically evaluate model performances, as even some of the most used models may have

unexpected flaws.

The first step of any model setup and performance evaluation is its calibration. Tools for the automatic calibration of the
model's parameters have been recently introduced, allowing for extending the search of the optimal values in a broader way.
The use of surrogate models accelerates global optimization. Analyses of how these tools can provide support and foster
a better comprehension of the numerical results are becoming increasingly frequent. For instance, examples are numerous

for the Delft3D model (in particular the FLOW module), one of the most popular models for simulating hydrodynamics of
natural environments. Xia et al. (2021, 2022) developed the PODS tool (Parallel Optimization with Dynamic coordinate search
using Surrogates); Schwindt et al. (2022) assessed the uncertainty of mixing-related model parameters thorough a Bayesian
calibrator; Garcia et al. (2015) and Baracchini et al. (2020b) adopted derivative-free algorithm for nonlinear least squares (DUD
from Ralston and Jennrich (1978)) within OpenDA (El Serafy et al., 2007) to optimize calibration and data assimilation in their

lake models. Many opportunities are offered by automated calibrators, saving time and finding global optimal solutions, and
many other optimization algorithms have been applied for the calibration of hydrodynamic models. We refer to Xia et al. (2021)
for a review of all these aspects. However, much care must be taken when model parameters are calibrated, either manually
or automatically. Many examples in literature indeed report optimal values laying in regions of the parameters space that hold
no physical relevance. Such unrealistic parameters are often justified as those giving the best model performance, but it is well

known that (i) different combination of parameters might give the same result and (ii) the performance of the model heavily
depend on what metric is adopted. This is particularly true for hydro-thermodynamic models calibrated on a single variable,
e.g. temperature profiles only (Amadori et al., 2021; Xia et al., 2022).

The objective of this work is to show how the use of automatic calibration can help identify flaws in the structure of
even state-of-the-art models. Analyses of how these tools can foster a better comprehension of the numerical results have

started to appear in the literature, e.g., the recent example of Bayesian calibration of a Delft3D model (Schwindt et al., 2022).
Here we adopt a similar approach to evaluate the parameterization of the heat distribution along the water column as coded
in Delft3D. In particular, we focus on the Secchi depth ($D_s$) parameter. The importance of properly accounting for water
transparency in hydro-thermodynamic models has been long recognized. Such a quantity enables to improve the prediction of
water temperature, eventually accounting for biologically-related mechanisms even if those are not explicitly parameterized





(e.g. reduction of transparency in occurrence with algal blooms, Rahaghi et al. (2024)). This aspect becomes even more relevant in data-scarcity contexts, where satellite-derived estimates of extinction coefficients can greatly improve the performance of even simplified models (e.g. Zolfaghari et al., 2017).

$D_\text{s}$ is used to model the distribution of the incoming solar shortwave radiation from the surface to deeper layers depending on how deep it can penetrate the water column. Despite being often considered a calibration parameter of the model (e.g.
Wahl and Peeters, 2014; Soulignac et al., 2017; Piccioni et al., 2021; Xia et al., 2021), $D_\text{s}$ is actually a measurable quantity and can significantly vary in time. Experience from previous applications of Delft3D in different lakes shows however that the calibrated value of $D_\text{s}$ generally resides in the lower range of measured values (Wahl and Peeters, 2014; Soulignac et al., 2018), or it is even smaller than observed values (e.g. Amadori et al. (2020) and Piccolroaz et al. (2019) rescaled the observed Secchi depth by a factor of 0.5). Through the automatic calibrator DYNO-PODS (Xia et al., 2022), we extend the search of
the optimal $D_\text{s}$ in a broader parameter space and explore regions uncharted to manual calibration. Similarly to Schwindt et al. (2022), we demonstrate that this allows to identify a flaw in the heat flux parameterization of Delft3D. In addition to Schwindt et al. (2022), we show how this let us improve the physics behind such a widely used model with a small, yet significant, modification to its source code.

In the following sections, we first formulate the problem (Section 2). Here we summarize how Secchi depth is used in
Delft3D (2.1), how surface warming is generally simulated in models (2.2), what are the physical implications of the current Delft3D parameterization (2.3) and of the modified version we implemented (2.4). In section 3, we introduce the calibration strategy and tests. In Section 4, we first show the results of the original formulation of Delft3D heat fluxes scheme (4.1). We then present the results of the calibration tests pointing out the model's unexpected behavior (4.2). Finally, we show the gain in physical reliability achieved after our modification of the Delft3D source code (4.3).

## 75 2 Formulation of the problem

### 2.1 Use of Secchi depth in models

The three-dimensional model Delft3D-FLOW (Lesser et al., 2004) numerically solves the Reynolds Averaged Navier Stokes equations (RANS) under the Boussinesq and shallow-water assumptions. The horizontal velocity components and the water surface are solved by integrating continuity and horizontal momentum equations, while vertical velocities are computed from
the continuity equation. The transport of heat is modelled by an advection-diffusion equation, assuming no heat exchange at the lake bed.

In the heat equation module of the model, all the heat flux terms are estimated with empirical formulas. Here, we focus on the penetration of solar shortwave radiation, which is modelled with a reduction of the energy flux per unit area ($\text{W.m}^{-2}$) along the water column, as in the commonly used Beer law:

$H_\text{sw}(z) = H_\text{sw0} \exp\left(-\gamma z\right) ,$ (1)





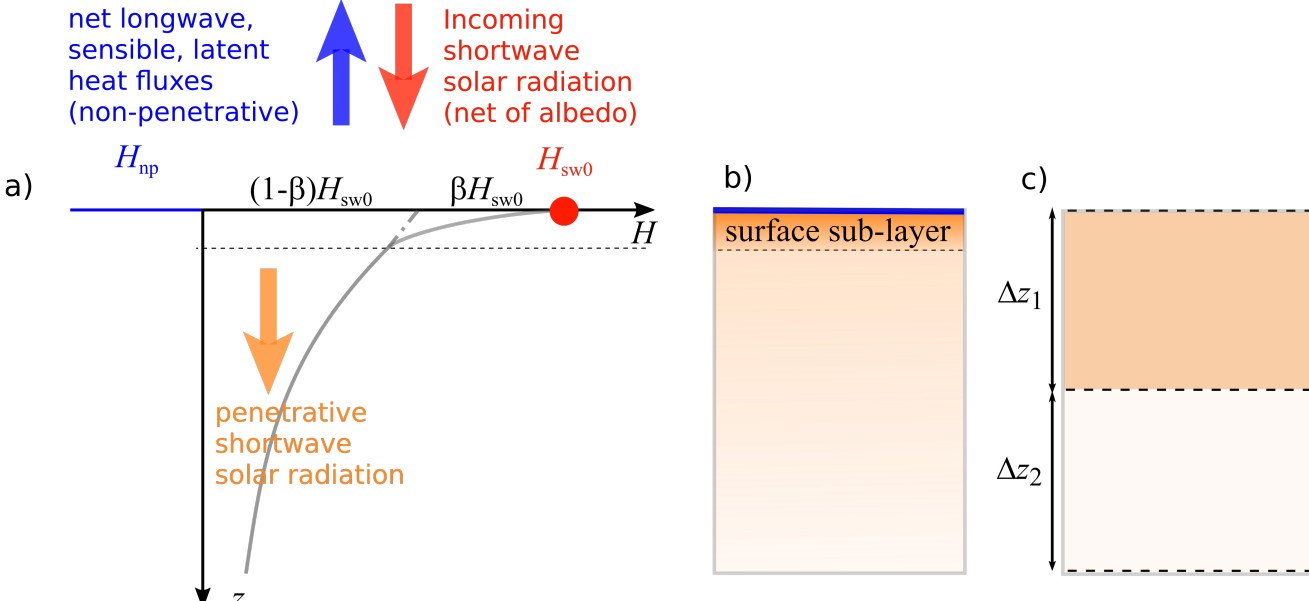

**Figure 1.** a) Radiative transfer at water surface. Non-penetrative terms $H_{np}$ are lost at the air-water interface (blue line, representative of the skin layer), while a sub-layer at the surface retains a fraction $\beta$ of incoming solar radiation $H_{sw0}$, and only the fraction $(1 - \beta) H_{sw0}$ of the solar shortwave radiation penetrates along the water column. The two grey lines represent the exponential decay of incoming heat flux in the surface sub-layer (above the horizontal dashed line) and beneath it. b) Conceptual illustration of how the process in a) evolves along the water column, with orange representing heat penetrating from the surface to the deeper layers; c) schematic of how a) is then parameterized in a vertically layered computational grid.

where $H_{sw0}$ is the downward shortwave radiation at the water surface (already considering the effect of albedo), $H_{sw}(z)$ is the energy flux that reaches the depth $z$, and $\gamma$ is the extinction coefficient ($\mathrm{m}^{-1}$). Such a simplified model refers to the light attenuation coefficient, which can be easily measured in-situ, and is appropriate for the the visible spectrum region (380-750 nm). Accordingly, the extinction coefficient $\gamma$ is defined based on the depth $D_s$ (m) at which the Secchi disk remains visible, through the simple relation:

$$\gamma = \frac{c_\gamma}{D_s}, \tag{2}$$

where the value $c_\gamma = 1.7$ is used as standard in Delft3D (Deltares, 2023).

However, the actual penetration of the shortwave radiation depends on the different wavelengths composing its spectrum. In principle, each wavelength has a different extinction coefficient (Zaneveld and Spinrad, 1980). Much of the shortwave radiation energy, in particular longer wavelengths in the near to shortwave-infrared region (750-2500 nm), is absorbed at the water surface regardless of its optical properties. The latter are indeed related to the concentration of constituents and mostly affect the extinction coefficient in the visible region (400-700 nm).





Among many formulations available to properly describe the penetration of downward solar radiation in the water column (Morel and Antoine, 1994), the simplified version reported in Henderson-Sellers (1986) is commonly applied in many

numerical models:

$$H_{\mathrm{sw}}(z) = (1 - \beta) \, H_{\mathrm{sw0}} \exp\left(-\gamma \, z\right) . \tag{3}$$

In such a formulation, $\beta$ is the fraction of the shortwave radiation absorbed a region close to the surface.

As a consequence, the surface layer is warmed up by a localized total heat flux,

$$H_{\mathrm{surf}} = H_{\mathrm{np}} + \beta \, H_{\mathrm{sw0}} , \tag{4}$$

which is the sum of the absorbed shortwave radiation $\beta \, H_{\mathrm{sw0}}$ and of the other non-penetrative fluxes $H_{\mathrm{np}}$ (see Figure 1). The non-penetrative heat flux,

$$H_{\mathrm{np}} = H_{\mathrm{lw}}^{\downarrow} - H_{\mathrm{lw}}^{\uparrow} \pm H_{\mathrm{sens}} \pm H_{\mathrm{lat}} , \tag{5}$$

is the result of the net downward longwave radiative flux $H_{\mathrm{lw}}^{\downarrow}$ (from the sky to the lake), the upward longwave radiation $H_{\mathrm{lw}}^{\uparrow}$ (emitted by the lake surface), and the sensible and latent heat fluxes, $H_{\mathrm{sens}}$ and $H_{\mathrm{lat}}$ respectively. All these terms are associated

to a generic surface $H_{\mathrm{surf}}$, but it is worth noting that non penetrative terms act just at the interface between water and air, while $\beta \, H_{\mathrm{sw0}}$ is absorbed in a layer of the order of tens of centimeters (Henderson-Sellers, 1986) (Fig. 1b). From the viewpoint of numerical modelling (Fig. 1c), the fraction $\beta$ of the incoming solar radiation represents a source of heat in a shallow layer of water, while non-penetrative terms (usually negative) represent a sink of heat. All these fluxes are normally parameterized in only one computational cell, i.e. the one immediately below the water surface, where the contributions of both equations 3 and

4 should be accounted for.

Modified versions of the Beer law were introduced in some lake-dedicated models to account for the different absorption of heat depending on the spectral bands of solar radiation. This is the case of the General Lake Model (Hipsey et al., 2019), where the authors attribute 55% of the incident solar radiation to near infra-red (NIR) and ultraviolet radiation (UVA,UVB) heating the surface directly. A default value of 45% for $\beta$ is implemented in CE-QUAL-W2 (Cole and Wells, 2015) but different

values ranging from 24 to 69% are recommended depending on the type of environment, with larger values attributed to pure (63%) and coastal (69%) waters, and smaller values (24 to 58%) to lake waters. A similar approach with different proportions (35% for NIR, 65% for PAR,UVA,UVB) was adopted by Thiery et al. (2014) to simulate Lake Kivu thermal structure with an ensemble of different lake models. Amongst these, the models explicitly including $\beta$ are SimStrat (Goudsmit et al., 2002), LAKEoneD (Joehnk and Umlauf, 2001), LAKE (Stepanenko and Lykossov, 2005), MINLAKE96 (Fang and Stefan, 1996).

Accounting for this fraction of heat at the surface is important in lakes, nevertheless the absence of this term in models like Delft3D never raised any concern in its past applications. The reason is that the extinction coefficient $\gamma$ estimated with equation (2), or equivalently the Secchi disk depth $D_{\mathrm{s}}$, is considered a calibration parameter, even if $D_{\mathrm{s}}$ is a measurable quantity.





## 2.2 General formulation of surface layer warming

Hydro-thermodynamic models like Delft3D solve a thermal energy balance, where the heat source is usually dominated by
the heat exchanged at the air-water interface. The relevant terms in the equation for the transport of heat can be represented as
follows:

$$\frac{\partial T}{\partial t} + \text{advection} + \text{diffusion} = -\frac{\partial \phi}{\partial z}, \tag{6}$$

where $T$ is water temperature, and we assume that the vertical coordinate $z$ is pointing downwards, so also the heat flux is
positive downwards (consistent with the direction of solar radiation). The source term on the right hand side of the equation
is responsible for heating the infinitesimal water volume locally. Incidentally, we note that the advection and diffusion terms
conserve the total heat content in the lake if no flux is exchanged at the boundaries.

In equation (6), we have simplified the notation by introducing the flux (units $\text{K.s}^{-1}.\text{m}$)

$$\phi = \frac{H}{\rho \, c_p}, \tag{7}$$

which scales the total heat flux $H$ with water's density $\rho$ (units $\text{kg.m}^{-3}$) and heat capacity at constant pressure $c_p$ (units
$\text{J.kg}^{-1}.\text{K}^{-1}$), assuming them to be constant, as a first approximation. Hereafter, the definition of $\phi_x$ based on equation (7) is
adopted for all heat fluxes $H_x$, where the subscript $x$ indicates the component.

The net heat flux at the air-water interface is

$$\phi_{\text{net}} = \frac{H_{\text{net}}}{\rho \, c_p} = \phi_{\text{sw0}} + \phi_{\text{np}} = (1 - \beta) \, \phi_{\text{sw0}} + \phi_{\text{surf}}. \tag{8}$$

In a depth-averaged model, the net heat flux is responsible for the change of the whole-lake temperature. Neglecting the sub-
daily variability, the warming of the lake is related to the total heat exchanged during a day:

$$\int_{\text{diel}} \phi_{\text{net}} \, dt = \int_{\text{diel}} \phi_{\text{np}} \, dt + \int_{\text{sunlight}} \phi_{\text{sw0}} \, dt, \tag{9}$$

where the integrals are defined on a 24-hour period (covering the periods of sunlight, with $\phi_{\text{sw0}} > 0$, and night, with $\phi_{\text{sw0}} = 0$).
The total exchanged heat, $\int_{\text{diel}} \phi_{\text{net}} \, dt$, is typically much smaller than the two individual components on the right hand side of
equation (9).

In conditions of approximate thermal equilibrium (e.g., Schmid et al., 2014), it is widely assumed that the lake temperature is
determined by a balance between all the fluxes exchanged at the surface. In these conditions it can be set that $\int_{\text{diel}} \phi_{\text{net}} \, dt \simeq 0$.
Therefore, the non-penetrative heat lost through the surface must approximately balance the solar heating, i.e., $\int_{\text{diel}} \phi_{\text{np}} \, dt \simeq$
$-\int_{\text{sunlight}} \phi_{\text{sw0}} \, dt$, so that the non-penetrative components of the heat flux have a cooling effect in most cases. In terms of diel
averaged values (indicated using angle brackets), the condition can be written as

$$\langle \phi_{\text{np}} \rangle \simeq -\frac{1}{\tau_{\text{day}}} \int_{\text{sunlight}} \phi_{\text{sw0}} \, dt < 0, \tag{10}$$





where $\tau_{\mathrm{day}}$ is 24 hours. Since the sub-daily variability of the non-penetrative fluxes is small compared to the solar radiation flux, it follows that $\phi_{\mathrm{np}} \simeq \langle \phi_{\mathrm{np}} \rangle$ (see, e.g., the example in Figure 2), and equation (8) implies that lakes are typically subject to (relative) cooling during night and to (relative) warming during sunlight.

The numerical model solves the discretized form of the heat equation (6) in a layer $\Delta z_i$. Neglecting the advective and diffusive fluxes for this simplified example, it reads

$$\frac{\Delta T_i^*}{\Delta t} = \frac{\phi_{i-1/2} - \phi_{i+1/2}}{\Delta z_i} = \sigma_i, \tag{11}$$

whereby we define the warming rate $\sigma_i$ (units K.s$^{-1}$). Note that $T^*$ here refers to the theoretical temperature for a lake where advective and diffusive terms are neglected. In fact, the temperature of the computational cell $i$ (the vertical index, assuming that the horizontal dimensions of the grid do not change vertically) increases if the heat flux entering from the upper face $(i-1/2$, for a staggered Arakawa C-grid) is larger than the flux leaving the cell from the lower face $(i+1/2)$.

Indicating with $i = 1$ the computational cell at the top of the water column (see also Figure 1c), the flux at the upper boundary condition of the model is $\phi_{1/2} = \phi_{\mathrm{net}}$, while the flux at the bottom of the first computational cell is $\phi_{3/2} = (1 - \beta)\phi_{\mathrm{sw0}} \exp(-\gamma \Delta z_1)$. Hence, the warming rate in the uppermost computational layer in equation (12) can be obtained by combining equations (3),(4), (7),(8) into equation (11):

$$\begin{aligned}
\sigma_1 \Delta z_1 &= (\phi_{\mathrm{sw0}} + \phi_{\mathrm{np}}) - (1-\beta)\phi_{\mathrm{sw0}} \exp(-\gamma \Delta z_1) \\
&= \beta \phi_{\mathrm{sw0}} + \phi_{\mathrm{np}} + (1-\beta)\phi_{\mathrm{sw0}} [1 - \exp(-\gamma \Delta z_1)].
\end{aligned} \tag{12}$$

Extending the same approach to the layers below the one at the surface, we can express the warming in layer $i = 2$ as

$$\begin{aligned}
\sigma_2 \Delta z_2 &= (1-\beta)\phi_{\mathrm{sw0}} \exp(-\gamma \Delta z_1) - (1-\beta)\phi_{\mathrm{sw0}} \exp[-\gamma(\Delta z_1 + \Delta z_2)] \\
&= (1-\beta)\phi_{\mathrm{sw0}} \exp(-\gamma \Delta z_1)[1 - \exp(-\gamma \Delta z_2)].
\end{aligned} \tag{13}$$

During sunlight ($\phi_{\mathrm{sw0}} > 0$), the second term on the right hand side of equation (13) is smaller than the first one, $(\Delta z_1 + \Delta z_2 > \Delta z_1)$. Hence, the layer $i = 2$ is always warmed. Similar considerations apply to all the other layers with $i > 1$. On the contrary, $\sigma_1$ may be negative (producing local cooling of the surface layer) even during warming periods, depending on the value of $\beta$ and $\phi_{\mathrm{np}}$.

The difference in warming rates of layers 1 and 2 as predicted by equation (12) and (13) may produce a vertical temperature gradient, which tends to be balanced by the vertical diffusive flux in equation (6). The effectiveness of such a flux in balancing the differential warming highly depends on the intensity of turbulence. In particular, increasing the eddy diffusivity could make the entire profile more homogeneous. As we will discuss in the following sections, turbulence might compensate the effect of different schemes for the penetration of solar radiation, for instance if $\beta = 0$, as in the original version of Delft3D heat equation module.

## 2.3 Case with $\beta = 0$

The original version of Delft3D, which adopts the Beer law as in equation (1), can be described as a particular case of the scheme described above in which $\beta = 0$. In this case the warming rate of the top layer ($i = 1$) expressed by equation (14) can





be simplified as

$$\sigma_1 \Delta z_1 = \phi_{\mathrm{np}} + \phi_{\mathrm{sw0}} \left[1 - \exp\left(-\gamma \Delta z_1\right)\right]. \tag{14}$$

190  When $\gamma$ is small ($D_{\mathrm{s}}$ very large, i.e., very transparent water), the last term within square brackets tends to vanish, so that the warming of the surface layer is $\sigma_1 \simeq \phi_{\mathrm{np}}/\Delta z_1$, which is typically negative because $\phi_{\mathrm{np}} \simeq \langle \phi_{\mathrm{np}} \rangle < 0$, as discussed in section 2.2. Therefore, the layer $i = 1$ may cool down also when sunlight is present ($\phi_{\mathrm{sw0}} > 0$), while the lower layers are always warmed up by the penetrative shortwave radiation, as implied by equation (15, obtained by setting $\beta = 0$ in equation 13).

$$\sigma_2 \Delta z_2 \quad = \quad \phi_{\mathrm{sw0}} \exp\left(-\gamma \Delta z_1\right) \left[1 - \exp\left(-\gamma \Delta z_2\right)\right]. \tag{15}$$

195  ## 2.4  Case with $\beta > 0$

We modified the Delft3D code by substituting the Beer law (1) with the complete version (3), as reported in A. Hence, the scheme reported in equations (12) and (13) with $\beta > 0$ applies. Focusing on the warming of the top layer and repeating the same argument as for equation (14), in case of small $\gamma$ the last term of equation (12) vanishes and $\sigma_1 \simeq (\beta \phi_{\mathrm{sw0}} + \phi_{\mathrm{np}})/\Delta z_1$. Now the sum of the two terms is not necessarily negative, as $\beta \phi_{\mathrm{sw0}}$ can balance the negative contribution of $\phi_{\mathrm{np}}$, and the

200  surface layer is not forced to cool down (or not so much) during sunlight. A proper choice of the coefficient $\beta$ guarantees a physically sound calibration of the numerical model.

## 3  Material and methods

Lake Morat, Switzerland is chosen as a test site for our study. This is one of the alpine lakes included in the AlpLakes network[1] for which a calibrated setup of the Delft3D model is available. The details of the case study and the base Delft3D setup have

205  been explained in B.

### 3.1  Calibration strategy

For the calibration of the Delft3D model we adapted to our needs the procedure DYNO-PODS based on a parallel surrogate global optimization method (Xia et al., 2021, 2022), which allows for calibrating multiple parameters simultaneously. DYNO-PODS runs parallel simulations and finds the best result at each iteration based on a cost function $\epsilon_k$ to be minimised. We refer

210  to the DYNO-PODS documentation and in particular to Xia et al. (2022) for an exhaustive description of the surrogate model implemented in DYNO-PODS.

The purpose of optimization is to correclty reproduce measured water temperature in one single station. We therefore tested two objective functions: (i) an error function $\epsilon_T$ based on the temperatures measured along the entire profile, as in equation (16); (ii) an error function $\epsilon_{T0}$ based on the temperatures measured at the lake surface only, as in equation (17).

---

[1]https://www.alplakes.eawag.ch





The distributed error $\epsilon_T$ is estimated by the average root mean square deviation

$$\epsilon_T = \sqrt{\frac{1}{N_t} \sum_{j=1}^{N_t} \frac{1}{N_z} \sum_{i=1}^{N_z} \left(T_{i,j} - \widehat{T}_{i,j}\right)^2}, \tag{16}$$

where $N_t$ is the number of temporal profiles, $N_z$ is the number of measuring points along the vertical, $T_{i,j}$ is the observed temperature at the depth $z_i$ and time $t_j$, and $\widehat{T}_{i,j}$ is the corresponding value simulated by the model. If $\epsilon_T$ is chosen as objective function, the model is optimised towards the best representation of both surface and deep water temperatures.

The error at the surface $\epsilon_{T0}$ is estimated via the root mean square deviation of the surface temperature $T_0$:

$$\epsilon_{T0} = \sqrt{\frac{1}{N_t} \sum_{j=1}^{N_t} \left(T_{0,j} - \widehat{T}_{0,j}\right)^2}. \tag{17}$$

If $\epsilon_{T0}$ is chosen instead as objective function, the calibrated model provides the best representation of surface temperature, regardless of the performance on water temperature below.

## 3.2 Calibration tests

Starting from a base setup ($\mathcal{O}$ in Table 1, for more details see B), we tested how the performance of the model could be improved by calibrating the Secchi depth. To preserve the time-dependence of measured values of Secchi depth we introduced a parameter $\delta$ defined as follows:

$$\widehat{D_s} = \delta\, D_s. \tag{18}$$

We first calibrated $\delta$ with the original model ($\mathcal{O}_{cal}$ in Table 1). Two separate calibration tests were performed considering the two objective functions defined in equations (16)-(17) (only the first value is reported in the Table 1, for more results see Table 2 in Section 4.2).

As a second step, we tested the effect of introducing the coefficient $\beta$ in a modified version of the model, $\mathcal{M}_{cal}$, where all other parameters were maintained as in $\mathcal{O}$, and $\delta$ and $\beta$ were calibrated.

Finally, we run a new base calibration for the modified model $\mathcal{M}$, where the full set of parameters was optimized, including $\beta$.

All calibration tests were performed in those months when water transparency prominently influences the formation of a stratified thermal structure in dimictic lakes, i.e. the spring and summer months subsequent to winter mixing. The simulated period was from 20 March to 14 August 2019. Sensitivity tests on the model response to variations of Secchi depth during the cooling season (from August onward) confirmed the assumption that it is possible to effectively determine the optimal value of $\delta$ limiting the calibration to the period when stratification forms (not shown).

The simulation time for 5 months was around 11 hours on a High Performance Computing Cluster. In order to gather a satisfactory convergence of the automated calibration tests, we run 10 iterations of 12 parallel simulation, for a maximum value of 120 evaluations for each calibration test. The number of evaluations is in line with what Xia et al. (2021) found in terms of speed of convergence towards a solution similar (in performance) to that from manual calibration.





**Table 1.** Calibration tests setups with values of the main parameters (wind drag corrective coefficient $\alpha$, free convection coefficinet (CFrcon), horizontal eddy diffusivity and viscosity coefficients (Dicouv, Vicouv) and Ozmidov length scale (Xlo), scaling of Secchi depth ($\delta$) and fraction of shortwave solar radiation absorbed in the first layer ($\beta$). $\mathcal{O}$ identifies the original Delft3D model, while $\mathcal{M}$ the modified version including $\beta$. Numbers are reported only when calibrated in the test, and "n.c." stands for "not calibrated". For $\mathcal{M}_{cal}$ test, not calibrated parameters are the same as in $\mathcal{O}$. "n.a." stands for "not applicable" and only refers to the $\beta$ parameter which does not exist in $\mathcal{O}$ model version.

| Model | Test | $\alpha$ | Cfrcon | Dicouv | Vicouv | Xlo | $\delta$ | $\beta$ | $\epsilon_T$ |
|---|---|---|---|---|---|---|---|---|---|
| | | - | - | $\mathrm{m}^2\mathrm{s}^{-1}$ | $\mathrm{m}^2\mathrm{s}^{-1}$ | mm | - | - | $^\circ C$ |
| Original | $\mathcal{O}$ | 0.94 | 0.019 | 2.36 | 1.65 | 0.338 | n.c. | n.a. | 0.57 |
| Original | $\mathcal{O}_{cal}$ | n.c. | n.c. | n.c. | n.c | n.c. | 0.84 | n.a. | 0.49 |
| Modified | $\mathcal{M}_{cal}$ | n.c | n.c | n.c | n.c | n.c. | 1.00 | 0.32 | 0.47 |
| Modified | $\mathcal{M}$ | 0.98 | 0.004 | 1.84 | 1.28 | 0.118 | n.c. | 0.33 | 0.45 |

## 4 Results

### 4.1 Base setup model

In this section, we present the results of the calibration of the original model $\mathcal{O}$ (without $\beta$, see Table 1) and without any modification of the Secchi depth. This setup implicitly corresponds to $\beta = 0$ and $\delta = 1$. Figure 2a shows the heat balance terms computed at an hourly time scale at the lake-atmosphere interface in a summer day (17 July 2019), for which in-situ measurements of water temperature are available. As expected from the theoretical considerations in Section 2.2, the net heat flux is heavily affected by the shortwave solar radiation $H_{sw0}$, which however vanishes at night, producing a strong sub-daily variability that is extremely large compared to the other terms. Moreover, if the shortwave radiation is excluded from the overall balance $H_{net}$, the resulting net heat flux accounting only for the non penetrative terms ($H_{np}$, dashed line) is negative also during daylight hours.

By using the heat fluxes internally computed in the calibrated Delft3D model, we estimated the local warming rate of the surface layer according to equation (14), i.e. neglecting advection and diffusion. In this simplified scheme, the heat flux at the surface resulted to be almost always negative indicating a predominantly heat loss term in the heat equation of the surface layer. As shown in Figure 2b, where the warming rate $\sigma$ is plotted everyday at noon, this happens even in the warming periods (spring-summer), with a few exceptions during extremely calm (i.e. low wind) and sunny days. At layers beneath the surface, $\sigma$ (estimated locally with equation (15)) has instead positive sign. Shortwave radiation indeed penetrates the water column with decreasing warming rate as depth approaches the Secchi depth $D_s$. Hence, excluding advective and diffusive terms from the simulation as in equation (11), would result in continuous cooling of the surface layer and slight warming of deeper layers. Although this is not the case in the Delft3D model, which accounts for advective and diffusive fluxes, the tendency can be noted also in the numerical results, as it will be shown below (Figure 4).



**Figure 2.** a) Heat fluxes computed using the meteorological forcing and the simulated surface water temperature from the base setup model ($\mathcal{O}$ in Table 1) for Lake Morat on a sample summer day, i.e. 17 July 2019. b) Warming rates simulated along the water column calculated according to equations (14)-(15) using the flux calculated by model $\mathcal{O}$. Profiles are displayed at 12:00 UTC. The black line represents the measured Secchi depth ($D_s$) considered in the simulation, with black dots indicating the exact measurement day. Depths are limited to 10 m to improve visibility of the surface layer.





**Table 2.** Calibrated parameters $\delta$ and $\beta$ and corresponding objective functions. "n.a." stands for "not applicable".

| Test | parameter | | objective function | |
|------|-----------|-------|------|-------|
|      | name | value | name | value |
| $\mathcal{O}_{cal}$ | $\delta$ | 0.12 | $\epsilon_{T0}$ | 0.83 °C |
|  | $\beta$ | n.a. |  |  |
|  | $\delta$ | 0.84 | $\epsilon_T$ | 0.49 °C |
|  | $\beta$ | n.a. |  |  |
| $\mathcal{M}_{cal}$ | $\delta$ | 0.57 | $\epsilon_{T0}$ | 0.8 °C |
|  | $\beta$ | 0.32 |  |  |
|  | $\delta$ | 1.00 | $\epsilon_T$ | 0.47 °C |
|  | $\beta$ | 0.32 |  |  |

## 4.2 Optimal scaling of Secchi depth

The results of the calibration tests $\mathcal{O}_{cal}$ and $\mathcal{M}_{cal}$ for assessing the optimal scaling for Secchi depth are reported in Table 2. The search for the optimal value of $\delta$ depends on the objective function used in the calibration tests.

In the original model $\mathcal{O}_{cal}$ (Figure 3a-b), if the target of the calibration is the surface water temperature ($\epsilon_{T_0}$ in equation (17)), the optimal scaling for Secchi depth unrealistically converges to $\delta = 0.12$. Smaller values of $\delta$ lead to warmer surface temperature because $\gamma$ becomes very large (low transparency; see equation (2)) and a large portion of the shortwave solar radiation is absorbed in the surface layer. If the target is water temperature over the entire water column ($\epsilon_{T_z}$ in equation (16)), the optimal value is $\delta = 0.84$.

The same plots from $\mathcal{M}_{cal}$ tests (Figure 3d-f), suggest that water temperature can be simulated without the need of drastically minimizing $\delta$. The optimal $\delta$ converges to 0.57 and 1 for the two objective functions $\epsilon_{T_0}$ and $\epsilon_{T_z}$, respectively. The first value (0.57) will be commented in the discussion section. Interestingly, Figure 3e-f show that the value of $\beta$ converges in a physically meaningful range and the optimal value is 0.32 for both objective functions (Table 2), fully consistent with literature values.

### 4.3 Effect of $\beta$

To understand what actually changes when introducing the parameter $\beta$ in the heat flux parameterization, Figure 4 shows the comparison between the fully calibrated models $\mathcal{O}$ and $\mathcal{M}$ (Table 1). Panels a,b show the simulated temperature profiles on a sample summer day in the two optimal simulations minimizing $\epsilon_{T_z}$, with and without $\beta$. The two thermal profiles appear very similar and consistent with measurements. Both simulations correctly capture the thermocline depth and the temperature of the surface well-mixed layer (about 7 m thick). However, in the model $\mathcal{O}$ (panel a) the temperature of the surface layer (23.27 °C) is slightly colder than that at layers below (23.35°C), while the surface in-situ temperature (24.15°C) is the warmest. Conversely, the model with calibrated $\beta$, i.e. $\mathcal{M}$ (panel b), returns 24.50°C at the surface.





**Figure 3.** Convergence of different calibration tests and objective functions (a, c, e: $\epsilon_{T_0}$, error on surface temperature; b, d, f: $\epsilon_{T_z}$, error on temperature along the water column). a-b): Calibration test $\mathcal{O}_{cal}$ for optimal $\delta$ in the original model $\mathcal{O}$ ($\beta$ implicitly equals 0); c-f) calibration of both $\delta$ (c,d) and $\beta$ (e,f) in $\mathcal{M}_{cal}$ tests with the modified model. The blue and orange arrows in panel a) show the main expected directions of the optimization problem in the calibration of $\delta$ (a-d). The orange shaded rectangle in panels a-d highlights the range of realistic values of $\delta$, i.e. around 1. The pink shaded rectangle in panels e-f highlights the range of values reported for $\beta$ in the literature, i.e. 0.2-0.6.





Although the difference in temperature is relatively small between the two model versions, the physics behind such thermal profiles is actually different. In Figure 4c-e we compare the eddy diffusivity $D_z$ as simulated by the $k$-$\epsilon$ model in $\mathcal{O}$ and $\mathcal{M}$ in three selected days. In the original model version, stronger turbulent vertical diffusion is simulated in the epilimnion. The largest difference is visible during daylight hours, when the $\mathcal{O}$ model (panel c) simulates homogeneous diffusivity of the order of $10^{-4}$ m$^2$s$^{-1}$ in the upper 2 m, while surface mixing is strongly inhibited in the $\mathcal{M}$ model (panel d), with $D_z$ generally lower

than $10^{-6}$ m$^2$s$^{-1}$. Thus, the enhanced mixing in the epilimnion simulated by model $\mathcal{O}$ compensates the unrealistic distribution of the warming rate caused by the absence of $\beta$, which would produce cooling of the surface layer (as Figure 2a shows). This effect is visible in the whole simulated period (panel e), with overestimation of $D_z$ of the order of $10^{-4}$ m$^2$s$^{-1}$ during daylight hours and higher differences during the stratified period.

## 5   Discussion

The comparison between the model outputs obtained with the original ($\mathcal{O}$) and modified ($\mathcal{M}$) versions of Delft3D clearly highlights that accounting for shortwave absorption in a shallow layer close to the water surface improves the performance of the model and, more importantly, provides a more realistic description of the physical processes driving surface warming in lakes. The best value of the scaling parameter $\delta$ obtained with the modified model is 1 (see Table 1), which implies that the measured value of the Secchi depth in Lake Morat is a reliable input parameter for the model and does not necessarily require calibration.

The fact that the measured Secchi depth can be assumed as a physically meaningful quantity is a significant advantage in the model setup with respect to the common expectation of several Delft3D users that Secchi depth must be calibrated (Wahl and Peeters, 2014; Soulignac et al., 2017; Piccioni et al., 2021; Xia et al., 2021). Moreover, the value of $\beta$ estimated through the automatic calibration accurately falls within the range $\beta \in (0.3, 0.4)$ observed in different lake environments (e.g., 0.4 in Dake and Harleman, 1969) and suggested in previous modeling applications (Thiery et al., 2014; Cole and Wells, 2015; Schmid and

Köster, 2016).

    To what extent such a modification of the heat fluxes parameterization is necessary to improve the model performances can be discussed building on the model's errors reported in Tables 1 and 2. Our results indicate that the base calibration of the original model $\mathcal{O}$ (Table 1) can already give satisfactory results, with an error $\epsilon_T = 0.57$°C along the water column. Such a result is possible thanks to the automated calibration, which allows to explore a wider range of testing parameters

combination compared to manual procedure. The relatively low error also confirms the general reliability of the model with its original formulation as demonstrated by the numerous successful lake applications in different lakes (e.g. Amadori et al., 2021; Baracchini et al., 2020a; Soulignac et al., 2017, 2018; Wahl and Peeters, 2014; Chanudet et al., 2012; Dissanayake et al., 2019, among many others). By rescaling $D_s$ through the calibration of $\delta$ ($\mathcal{O}_{cal}$ test), the error reduced to $0.49$°C. Such a result is in line with the observations by Amadori et al. (2020); Piccolroaz et al. (2019), i.e. that for the original scheme to

perform better, reducing the measured Secchi depth is advised. An improvement of $0.08$°C in the model error along the water column is considerable (14%), but it is achieved at the expense of the physical meaning of Secchi depth information. If $\beta$ is



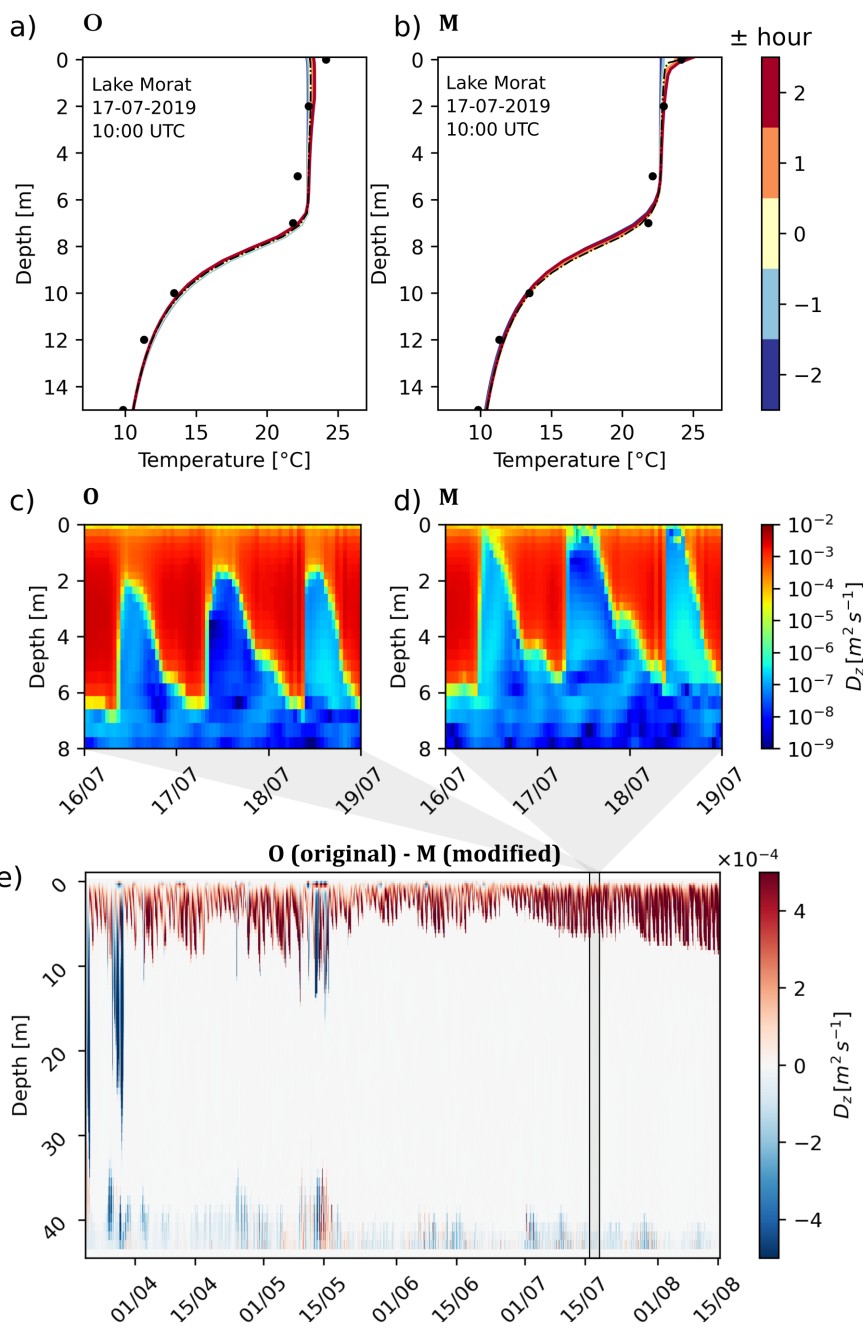

**Figure 4.** a-b) Temperature profiles simulated (lines) and measured (black dots) on a summer day (17 July 2019) in Lake Morat implementing the $\mathcal{O}$ and $\mathcal{M}$ model configurations with target $\epsilon_{T_z}$. Model results are displayed at the measurement time (10:00 am, black dashed line) and in a time range of $\pm 2$ hours around 10:00 am. c-d) Vertical eddy diffusivity simulated in the two model setups as in a-b) $\pm 2$ days around 17 July 2019 (black straight line in panel e). d) Difference in simulated eddy diffusivity between the two models shown on an hourly basis.





introduced and calibrated ($\mathcal{M}_{cal}$ test), we see that the final error $\epsilon_T = 0.47°$C is only slightly smaller than in the previous case, but significant gain is obtained in the physical consistency of all inputs ($\delta = 1.0$, $\beta = 0.32$).

The interesting element of our analysis is that the parameter calibration allows the original model to get close to reality
despite the faulty parameterization of the heat absorption at the surface. The price of such optimization is however the alteration of other physical processes that compensate the flaw in the model formulation. This is evident from the enhanced diffusivity at the surface in the original model formulation (Figure 4) which derived from an unrealistic instability between the upper layer and those below due to surface cooling. Such overestimation of surface mixing was also observed by Biemond et al. (2021), who compared Delft3D model results in Lake Garda with in-situ turbulent kinetic energy dissipation ($\varepsilon$) profiles. In particular,
such overestimation was stronger in stratified conditions and was present also when temperature profiles were appropriately simulated.

Layer thickness also plays a role in the optimal scaling of $\delta$. When surface water temperature is the calibration target ($\epsilon_{T_0}$ as objective function), the optimal value of $\delta$ is 0.57 (Figure 3c) even if $\beta$ is included in the parameterization and is calibrated. While 0.57 is definitely more acceptable than the value 0.12 obtained in the case without $\beta$ (3a), it is still significantly smaller
than the expected realistic value, i.e. $\delta = 1$. We speculate that this behavior can be interpreted by recalling that the layer in which the $\beta$ fraction is absorbed is generally considered to be $\sim 60$ cm thick (Henderson-Sellers, 1986). This thickness is used by several models as the reference depth where the exponential decay starts (Zaneveld and Spinrad, 1980; Piccolroaz et al., 2024). Since the surface layer thickness in our model is 0.26 m (Table B1), it is possible that the value of $\delta$ lower than expected (i.e. 0.57) is again compensating for trapping the fraction $\beta$ in a too thin layer. This suggests that including such reference
depth in the parameterization, and eventually calibrating it, might result in a more realistic value of $\delta$ also when optimizing the model to simulate surface water temperature only.

## 6  Conclusions

Automatic calibration tools are powerful for optimizing parameter selection in complex models, helping modelers in achieving realistic simulations of the studied environment. Here, we demonstrated that such tools can also be utilized by developers to
assess whether their implementations of models align with the physics, and by users to identify potential improvements needed in the modeling framework. While issues with lake modelling often stem from inappropriate meteorological conditions, users and developers can leverage these toolboxes to detect code-related issues.

We tested the response of a hydrodynamic model, Delft3D, which is widely used by the limnological community, to the task of simulating the vertical distribution of heat in lakes. The process is regulated by the well-known Beer law describing
the absorption rate of shortwave radiation along the water column, but its incomplete implementation in the numerical model requires a compensation mechanism (via a non-physical adjustment of the Secchi depth instead of the including the absoprtion of a fraction of solar shortwave radiation at the surface) to optimally simulate surface water temperature. By exploiting a recently developed calibration tool (DYNO-PODS), based on surrogate models, we were able to unveil such a limitation of the Delft3D parameterization that was never explicitly discussed, and to fix it with a small modification of the source code.





In conclusion, we recommend caution in blindly applying automatic tools, emphasizing the importance of evaluating the physical significance of the obtained calibration parameters. We believe that the extensive use of such calibrators, and the insightful analysis of the results of the optimization, may offer similar outcomes in other applications, even in the case of computationally heavy simulations.

**Appendix A:  Source code**

We report here the parts of the Delft3D code which were modified to include the effect of $\beta$ according to equation (3). Such modification is made in the subroutine `heatu.f90`.

At surface layer, the original source code lists:

```
qink  = corr * qsn * (1.0_fp - exp(extinc*zdown)) / extinc
```

We modified as:

```
qink  = corr * (1.0_fp-beta_sw) *qsn * (1.0_fp - exp(extinc*zdown)) / extinc
        + qsn * beta_sw
```

At deeper layers, the original source code lists:

```
qink  = corr * qsn * (exp(extinc*ztop) - exp(extinc*zdown))
```

We modified as:

```
qink  = corr * (1.0_fp-beta_sw) * qsn * (exp(extinc*ztop) - exp(extinc*zdown))
```

**Appendix B:  Test site and base model calibration**

Lake Morat (Lake Murten in German) (Fig. B1a,b) is a Swiss lake situated at 429 m above sea level with a surface area of 22.8 km$^2$, an average depth of 24 m, and a maximum depth of 45 m. In-situ water temperature profiles are sampled at monthly basis in the middle of the lake (Fig. B1b,c), together with measurements of water transparency (expressed as Secchi depth, Fig.
B1d).

The details on the model grid size and time spacing of the Delft3D model for this lake are reported in Table B1. As atmospheric forcing at the surface boundary of the lake we used COSMO-1 space and time varying meteorological variables (air temperature, relative humidity, wind speed, cloud cover, shortwave radiation and air pressure) provided by MeteoSwiss at hourly time resolution and  1.1 km spatial resolution. The assimilated outputs, i.e., reanalysis data, based on observational
measurements (Voudouri et al., 2017) were used for this purpose.

The base setup for this lake model was obtained from a year-long automated calibration implementing DYNO-PODS (Xia et al., 2022). The full set of parameters are reported in Table 1 in the main text and include the background horizontal viscosity and diffusivity coefficients (indicated as Vicouv, Dicouv in Delft3D, see Deltares 2023), the Ozmidov lenght scale (Xlo), the

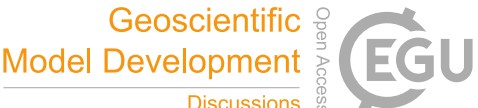

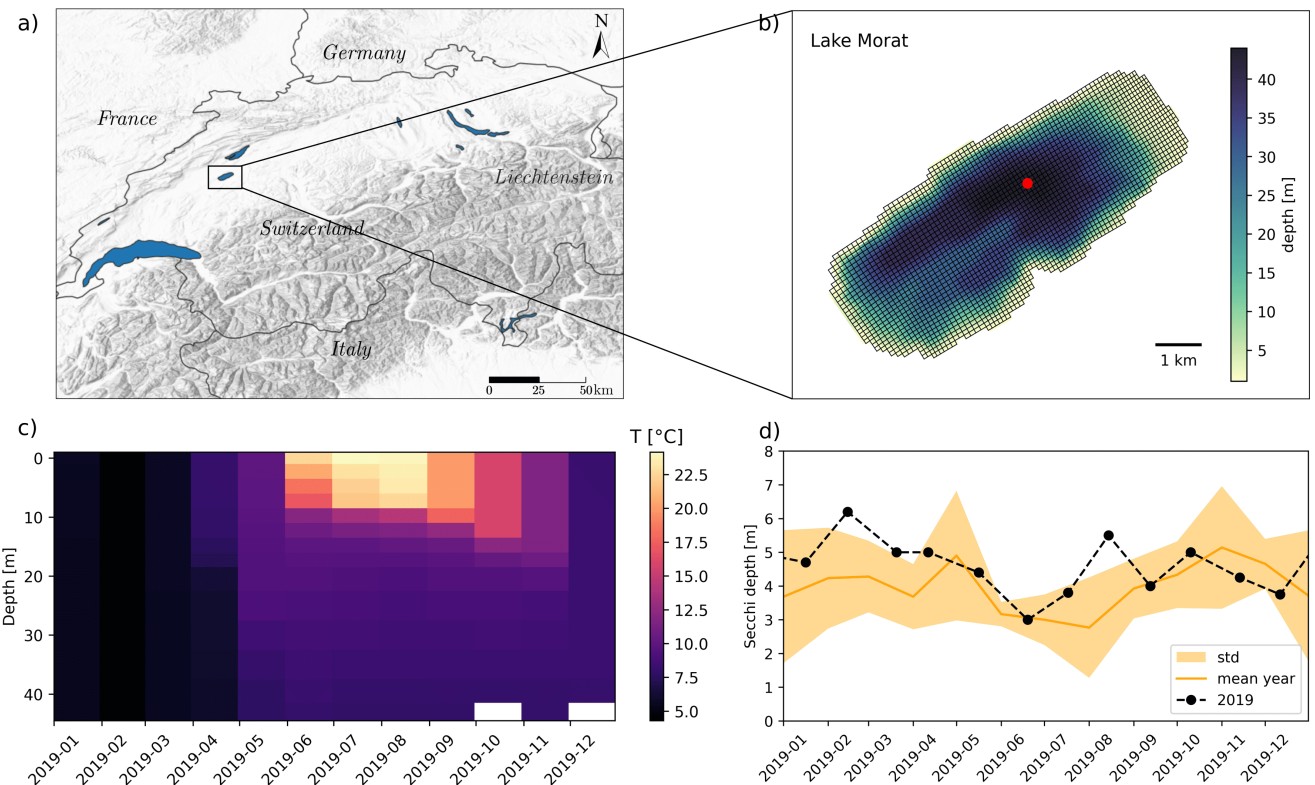

**Figure B1.** a) Location of Lake Morat in the alpine area; b) computational grid, bathymetry and location of in-situ monitoring station (red dot); c) heatmap of the monthly in-situ water temperature profiles for the year 2019 measured in the monitoring point; d) Time series of in-situ water transparency for 2019 (black dots) and mean year computed from the 2016-2021 time series (mean: orange thick line; standard deviation: orange shaded area).

**Table B1.** Base setup for the Delft3D model of Lake Morat.

| Nmax | Mmax | Kmax | $\Delta x, y$ | $\Delta z$ | $\Delta t$ |
|------|------|------|------|------|------|
| - | - | - | m | m | s |
| 41 | 88 | 77 | 73-110 | 0.26-1 | 60 |



free convection coefficient (Cfrcon), and a corrective coefficient on wind drag coefficient $\alpha$. For the wind drag coefficient,
as in recent applications of the Delft3D model to peri-alpine lakes (e.g. Amadori et al., 2021), we took as a reference the
parameterization proposed by Wüest and Lorke (2003) to set the coefficients of the piecewise function implemented in Delft3D:
$C_{dA}^0 = 0.0044$ for wind speed lower than $U_A = 0.5 \text{ m s}^{-1}$, $C_{dC}^0 = 0.002$ for values higher than $U_C = 10 \text{ m s}^{-1}$, and linear
interpolations between $C_{dB}^0 = 0.001$ and $C_{dA}^0$ and $C_{dC}^0$, respectively for wind speed between $U_B = 4.5 \text{ m s}^{-1}$ and $U_A$ and
$U_C$. The wind effect is also taken into account in the parameterization of forced latent and sensitive heat fluxes. Instead of
calibrating parameters like the "Stanton" and "Dalton" numbers, respectively for the sensible and latent heat fluxes (Deltares,
2023), we set the model to use $C_d$ also in calculating these heat fluxes.

To account for the uncertainties related to the input wind speed and to calibrate the wind function for the forced heat fluxes,
the drag coefficient was adjusted by introducing the corrective parameter $\alpha$ as follows:

$$C_{di} = \alpha C_{di}^0. \tag{B1}$$

Background values of the vertical eddy viscosity and diffusivity (Vicoww, Dicoww) were set as equal to molecular values
(i.e. $10^{-6}$ and $10^{-9}$ m$^2$s$^{-1}$, respectively), while all the other calibration parameters (e.g. bottom roughness) were set as default
values.

The Secchi disk depth $D_\text{s}$ was provided as time series with approximately monthly frequency, as obtained from in-situ
measurements. These values were not modified. As objective function for this base calibration, we used the error along the
entire water column $\epsilon_T$.

*Code availability.* The current version of the model is available at https://github.com/eawag-surface-waters-research/Delft3D/tree/d3d4/
research/surface_heat_transfer under the GPLv3 licence. The exact version of the model used to produce the results used in this paper is
archived on Zenodo (Amadori et al., 2024) as are input data and scripts to run the model and produce the plots for all the simulations
presented in this paper.

*Author contributions.* Conceptualization: M.A., M.T.; Data curation: M.A.; Formal analysis: M.A., M.T.; Funding acquisition: D.B., M.T.;
Investigation: M.A., A.I.R., D.B., M.T., Methodology: M.A., M.T., Software: M.A., M.T.; Validation: M.A.; Visualization: M.A., M.T.;
Writing – original draft preparation: M.A., M.T.; Writing – review & editing M.A., A.I.R., D.B., M.T..

*Competing interests.* The authors declare no competing interests

*Acknowledgements.* This work was supported by ESA through the grant AlpLakes (Contract No.4000136401; AO/1-8216/15/I-SBo; Sci-
entific Exploitation of Operational Missions - SEOM S2_4SCI Land and Water Coastal and Inland Waters). MT acknowledges the support





of the project DICAM-EXC (Departments of Excellence 2023–2027, Grant L232/2016), funded by by the Italian Ministry of Universities and Research (MUR). The authors are grateful to Wei Xia for his prompt assistance on the use of DYNO-PODS and to Claudia Giardino, Mariano Bresciani, Sebastiano Piccolroaz, Daniel Odermatt, Fabian Bärenbold and Jorrit Mesman for insightful discussions and literature suggestions.



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
