# Peer review of "Using automatic calibration to improve the physics behind complex numerical models: An example from a 3D lake model using Delft3D (v6.02.10) and DYNO-PODS (v1.0)"

_Geoscientific Model Development, 2024_

## Author Comment (AC1)

Response to Reviewer1 - 'Comment on gmd-2024-118', Andrea Fenocchi

This paper reports a very witty use of automatic calibration algorithms to highlight an unphysical flaw in surface heat flux parameterisation in the widely used Delft3D hydrodynamic model. Such issue is often not evident in common uses of the model, in which temperature calibration is performed over the full water depth and calibrated turbulence parameters compensate the error. The issue was thus tracked down optimising surface temperature only and blocking turbulence parameter calibration, taking benefit of the optimal solutions found through unbiased automatic calibration. A correction in the shortwave radiation absorption parameterisation is then formulated and included into the model, making it able to reproduce the micro-stratification often observed in the surface-most part of the epilimnion of open waters.

The research is well developed and mostly efficiently conveyed in the manuscript. The topic is relevant to the journal and the results are useful to the scientific community. I'm highlighting below some prompts for further discussion of results and of their implications. I'm also suggesting a careful final proofreading of the paper aimed at optimising general clarity.

Given all the above considerations, I think this paper can be published after minor revisions.

We are pleased to read that our manuscript has raised the reviewer's interest and we sincerely thank him for his thorough review. The reviewer has fully grasped the essence of our work. Our message is not that we improved the heat fluxes parameterization of Delft3D, but rather that we identified the oversimplification of the surface layer heat penetration scheme by using optimization tools with critical thinking.

We addressed all reviewer's comments and we improved the clarity of our manuscript by revising the English form as he suggested. In the following reply, our response to the reviewer's comments is written in blue, **new text in the revised paper is formatted in italic bold,** *text from the original manuscript in italic*. We will refer to line numbers reported in the track-changes document.

SPECIFIC AND TECHNICAL COMMENTS:

L32: "allowing for extending" is not proper English

We modified it with "*enabling a broader and more efficient search for the optimal parameters*" (lines 36-37).

L33: the meaning of "surrogate models" should be disclosed here

We included the following lines 47-52:

**"A surrogate model, also known as emulator, is a simplified and computationally efficient empirical model (Castelletti et al., 2010) that mimics the behavior of a computationally expensive model based on real model training data. In the search of the optimal parameters,**

*the largest computation cost is indeed related to the real model runs. Thus, fast surrogate model runs are alternated to real model runs such that a lower number of real model evaluations is needed. Surrogate models can be included in traditional optimization tools such as Bayesian calibration (Ma et al., 2024) and are highly effective in speeding up the calibration process."*

L55: "in occurrence with" is not proper English

We removed the expression from the paper.

L77-78: Delft3D-FLOW solves the shallow-water equations when used in 2D mode, the RANS equations with the Boussinesq assumption when used in 3D mode. This should be better specified.

The shallow-water approximation actually applies to the governing equations of Delf3D-FLOW also in the case of 3D mode. As Lesser et al. (2004) say:

"The DELFT3D-FLOW module solves the unsteady shallow-water equations in two (depth-averaged) or three dimensions. The system of equations consists of the horizontal momentum equations, the continuity equation, the transport equation, and a turbulence closure model. The vertical momentum equation is reduced to the hydrostatic pressure relation as vertical accelerations are assumed to be small compared to gravitational acceleration and are not taken into account. This makes the DELFT3D-FLOW model suitable for predicting the flow in shallow seas, coastal areas, estuaries, lagoons, rivers, and lakes. It aims to model flow phenomena of which the horizontal length and time scales are significantly larger than the vertical scales."

What the authors do not explicitly say in this paragraph, but clarify later on in eq.15 of the same paper, is that the vertical velocity is computed from the continuity equation. The authors indeed write:
"The vertical velocity, $\omega$, in the $\sigma$-coordinate system, is computed from the continuity equation". The same applies, as far as we know, to the case of the z-coordinate system.

We think that providing these details deviates the focus of the paper from the main objective and we therefore did not modify the main text.

L78-80: if I recall correctly, the vertical momentum equation is still present, yet it is simplified to a hydrostatic equilibrium condition

See the reply above.

L92: the c_gamma = 1.7 factor actually dates back to the pioneering work of Poole & Atkins (Poole, H.H., Atkins, W.R.G., 1929. Photo-electric measurements of submarine illumination through-out the year. J. Mar. Biol. Ass. U.K. 16, 297-324)

Thanks, we added this reference as follows (line 119):

*"where the value cγ = 1.7 is used as standard in Delft3D (Deltares, 2023) and was originally proposed by Poole and Atkins (1929)"*

L95: "in the near" is not proper English

"Near" here is intended as near-infrared (NIR range 750 nm to 1400 nm) to be distinguished from shortwave-infrared (SWIR range 1400-2500 nm). We understand the original phrasing could be misleading and we improved the sentence as follows (lines 123-124):

*"in particular longer wavelengths in the near-infrared to shortwave-infrared region 750-2500 nm"*

L102: add "in"

Thanks for spotting this typo. We corrected the text.

L132-134: define "phi" at first occurrence

We modified as follows (lines 166):
*"where T is water temperature and $\phi$ is the vertical heat flux."*

L135-136: although I get the meaning of this sentence, it should be clarified

We revised it as follows (lines 167-171):
*"The source term on the right-hand side of the equation represents the local heating, which depends on the absorption of the radiative heat flux. Without this source term, the advection and diffusion terms can only redistribute the heat in the domain, so that the total heat content of the lake is conserved if no flux is exchanged at the boundaries."*

L139: remove "'s"

Done, thanks.

L140: use "i" as generic subscript for added clarity

We prefer keeping "i" as subscript for the vertical layers (see eq. 11).

L163-164: improve the definition of "T*"

We modified as follows (lines 195-205):
*"The numerical model solves the discretized form of the heat equation (6) in an $i - th$ layer with the thickness of $\Delta z_i$. If the advective and diffusive fluxes are neglected, the rate of change of the temperature $T^*_i$ (where * denotes the idealized case of no advection and diffusion) can be written as:*

*eq.(11)*

*whereby we define the warming rate $\sigma_i$ (units $K\ s^{-1}$). In fact, assuming that the horizontal dimensions of the grid remain vertically constant and in the absence of advection and diffusion, the temperature $T^*_i$ of the computational cell i increases only if the heat flux entering the cell exceeds the flux leaving the cell. When the heat source comes from above, this means that warming occurs if the heat flux entering from the upper face $(i - 1/2$, for a staggered Arakawa C-grid) is larger than the flux leaving the cell from the lower face $(i + 1/2)$."*

L187: replace "(14)" with "(12)"

Suggestion followed, thanks.

L194: ok, but continuing your reasoning, also Eq. 15 drops to zero if D_s is very large. I get the meaning, but the reasoning and the relevant outcomes of this passage should be better explained

We thank the reviewer for pointing this out. It is true, in asymptotic terms, that also equation 15 drops to zero for $\gamma \to 0$. We slightly modified the paragraph as follows (lines 230-236):

*"When γ is small (Ds very large, i.e., very transparent water), the last term within square brackets becomes less important and non-penetrative terms prevail. As discussed in section 2.2, φnp ≃ ⟨φnp⟩ < 0 (see also Figure 2a). Therefore, the layer i = 1 may cool down also when sunlight is present (φsw0 > 0).The lower layers instead always receive heat, despite being maybe small in very transparent conditions, as the warming rate σ2 only depends on penetrative shortwave radiation as implied by equation (15), which is obtained by setting β = 0 in equation 13."*

L228: specify that D_s with the hat is the one used in the model and the one without hat is the one from field data.

We have added the following sentence after equation (18), line 273:

*"where $\widehat{D_s}$ is the Secchi depth value used in the model and $D_s$ refers to the original value measured on-site."*

L230: explain why you are reporting only the first values in Table 1.

Table 1 is intended as a recap of the calibration tests we run for obtaining an optimized set of parameters by using an objective function that is standard practice in numerical modeling. We consider the error along the entire water column ($\epsilon_T$) as a standard objective function. The values reported clearly show a progressive improvement in the model performance and in the physical reliability of the parameters from $O$ to $M$: $\epsilon_T$ reduces, $\delta$ goes to 1, $\beta$ ranges around 0.32, and mixing expressed by horizontal turbulent coefficients slightly reduces.

Table 2 has another scope, which is showing the impact of the different objective functions and the unphysical results obtained with the original model O_cal if $\epsilon_{T0}$ is minimized. We believe that separating the results in two different tables helps the reader understand the methodological scope of Table 1. We better specified this in lines 276-277:

**"Since the error along the entire water column (εT) is a standard objective function, only the details of these results are reported in Table 1. However, we also show and discuss the relevant *results of tests with εT_0 below in Table 2 in Section 4.2)."***

L243-244: what do you mean by "speed of convergence"?

We mean the number of model evaluations needed to achieve a model performance comparable to manual calibration. We modified the sentence as follows (lines 290-291):
*"This aligns with Xia et al. (2021), who found that 120 model evaluations were necessary to achieve a solution similar (in performance) to manual calibration in their case study."*

L282-284: ok, but surface temperature measurements are always somehow elusive, especially under no-wind or weak wind conditions for which surface micro-stratification builds up, as there

is no consolidated methodology on "how deep should I plunge down the thermistor to make a surface temperature measurement?". This could be discussed.

We agree, thanks for pointing this out. The near-surface value of water temperature is sensitive to many factors: the plunging depth of the instrument, water level fluctuations (especially relevant for permanent thermistor chains), and exposure of the near-surface sensor to direct sunlight. Any alteration of in-situ surface water temperature measurement has potentially large impacts on the simulation of heat fluxes in a model tuned to reach that temperature. In the lines indicated we already mention that the difference in temperature between the two model versions and the measured value is subtle and not worth attention per se. What is interesting is how the two models achieve that result: the original model O-cal amplifies mixing at the surface to compensate for the unphysical cooling caused by the inaccurate parameterization. We included a paragraph in the Discussion section to address this aspect. *See lines 404-410:*

*"We have shown in Figure 4 that the largest difference between the two model configurations is at the surface, and in particular during daylight hours in the stratified period. In the temperature profiles shown, a strong gradient is present between the near-surface temperature (i.e. 0 m) and the layer immediately below (2 m). On other days, when the measured surface temperature is relatively well mixed with the layers below, the difference between the two model versions is not that relevant. In fact, if enough mixing is physically provided by, for example, strong wind-induced turbulence, the model does not need to generate artificial convection to compensate for the negative buoyancy at the surface caused by the incorrect parametrization."*

and later on in lines 424-429:

*" When evaluating the model performance based on water temperature just below the surface, it is worth recalling that the in situ observation of such temperature is sensitive to many factors: from the plunging depth of the instrument, to water level fluctuations modifying the actual water depth of e.g. permanent thermistor chains, to to the exposure of the near-surface sensor to direct sunlight (Bärenbold et al., 2024). In this regard, only detailed vertical resolution of in situ measurements can support understanding of how well the surface temperature microstructure is reproduced by the model."*

L300-302: ok, but there are other reasons for which the Secchi depth parameter could still be calibrated, such as to lump intrusions by tributaries, baroclinic mixing effects not simulated by the model due to the Boussinesq assumption, imperfect turbulence modelling, imperfect atmospheric boundary forcing. This could be discussed.

We included the following paragraph discussing this aspect in lines 363-376 and we thank the reviewer for the suggestion:

*"Systematic calibration of model parameters, including Secchi depth, is essential not only for more accurately representing the physics of the system but also for addressing the imperfections inherent in the model (e.g., neglected baroclinic mixing, imperfect turbulent schemes, unaccounted effects of tributary intrusions, inaccurate atmospheric forcing). Among all the calibration parameters, the Secchi depth is perhaps the most frequently monitored and is one of the easiest to measure, both in situ and through remote sensing. However, several*

*previous modeling studies have highlighted significant deviations of this parameter in Delft3D from the measured values. Here, we demonstrate that this discrepancy is due to a misimplementation of certain physical phenomena in the Delft3D model, which is supported by the deviation of calibrated Secchi depths after correcting for it. However, there may still be a need to calibrate Secchi depth to account for potential errors in measurements or the sparse resolution of Secchi depth data in both time and space. Meanwhile, calibration of other parameters (e.g., wind drag, background eddy coefficients) will remain necessary to compensate for unresolved processes in the model. We believe that it is good modeling practice to avoid compensating for inaccuracies in one process by adjusting the parameters of others. We therefore suggest focusing on improving model deficiencies and promoting the correct use of parameters to ensure more reliable and physically sound results."*

L319-326: I think that this is the most relevant result of this work, the fact that by improving the model you are able to reproduce the typically observed micro-stratification in the top-most part of the surface mixed layer (Figure 4b). You should try to make this achieving more evident, starting from the abstract.

We agree that this is an important result. However, as reviewer 2 pointed out clearly, and as reviewer 1 also mentions in the next comment (L334-336), other models are already providing a reliable reconstruction of the surface micro-stratification thanks to more sophisticated parameterization of surface heating. We consider Fig. 3 as the key result of this contribution, which aims at promoting a smarter use of automated calibration.

L334-336: if I got it right, this is close to what the GOTM (General Ocean Turbulence Model) 1D model already does, following the Paulson and Simpson parameterisation (Paulson, C. A., Simpson, J. J., 1977. Irradiance measurements in the upper ocean, J. Phys. Oceanogr., 7, 952-956.), in which also the near-infrared shortwave radiation has an exponential decay, yet over a much shallower layer than the ordinary shortwave radiation. This could be discussed.

Thanks for this comment. What we propose is indeed conceptually similar to what GOTM does, but simpler in the parameterization. We indeed only suggest that the number of layers of the model where a larger absorption of shortwave radiation occurs depends on the layer thickness and could be calibrated as well. This could be done simply by distributing the $\beta$ fraction of shortwave radiation in a given number of layers, or by setting a reference depth. For the latter case, a solution like that of GOTM would be preferable, as these models allow setting different extinction coefficients for the different portions of the spectrum of the solar radiation. The GOTM default is the 2-band model following Jerlov (1968) "Optical Oceanography" and Paulson and Simpson (1977). More sophisticated versions also include a 9-band model, following Paulson and Simpson (1981), 10.1029/JC086iC11p11044 (see e.g. Karagali et al., 2017, 10.1002/2016JC012542). We added reference to GOTM and to other models as follows (lines 431-434):

*"However, we expect the best improvement to come with schemes similar to those already implemented in GOTM (General Ocean Turbulence Model, Burchard et al. (1999)), ROMS (Shchepetkin and McWilliams, 2005) or MITgcm (Marshall et al., 1997), which allow different extinction coefficients for different portions of the solar radiation spectrum, following Paulson and Simpson (1977)."*

L338-342: it should be pointed out here that automatic calibration is (mostly) unbiased, as opposed to user-sensitive manual calibration, and as such can better highlight objective flaws in models

Thanks for this interesting suggestion. We modified the sentence as follows (lines 449-453): *"Unlike manual calibration, which is often user-sensitive and prone to bias, automatic calibration is objective and allows to more effectively highlight inherent flaws in model formulations. While issues in lake modeling often arise from inadequate meteorological inputs, automatic calibration can help users and developers detect code-related issues, offering a more unbiased, systematic approach."*

This is an overall well-written and interesting manuscript demonstrating how optimization toolboxes can help uncover limitations in simplified models/parameterizations within complex numerical models, with an application to a widely used 3D hydrodynamic model. The authors show that in the present version of this hydrodynamic model, the parameters of the shortwave radiation module often must be tuned to unphysical values to achieve the best response in lake temperature. They convincingly explain that this is due to a simplification in the shortwave radiation penetration model and provide a physically based improvement to the code.

In general, I find this an interesting study worth publishing. However, some context is missing, and the wording should be adjusted in some places. The authors demonstrate for one, no doubt interesting and relevant case, that their framework can work well, providing a simple way to improve a model with a few lines of code. However, it is my understanding that other widely used 3D numerical hydrodynamic models already have a better implementation of the shortwave radiation penetration, e.g., those based on Jerlov water types, as implemented in the MITgcm and ROMS (see Paulson & Simpson, 1977; https://doi.org/10.1175/1520-0485(1977)007%3C0952:IMITUO%3E2.0.CO;2). Thus, the improvements the authors suggest are no revolution as such.

We thank the reviewer for the thoughtful analysis and the detailed comments. However, we believe that the main message of our work is different. It is not questionable that models like Delft3D are limited by inherent assumptions and that other models in literature sometimes provide better results and have more accurate parameterizations. The focus of our manuscript is not that we improved a few lines of code in Delft3D, but that we identified which lines needed to be improved by interpreting the results of an automatic calibrator. Recent optimization toolboxes, using surrogate models, enable efficient parameter calibration, saving time and computational resources. These optimizations sometimes yield physically meaningless parameter values. In these cases, modelers face the decision of whether to use such unrealistic values or try to understand why such values seem to be the optimal ones. Unfortunately, the first option is the most popular one. Here we showed how rewarding it can be if the second is chosen.

We addressed all reviewer's comments and we improved the clarity of our manuscript by revising the English form as he suggested. In the following reply, our response to the reviewer's comments is written in blue, **new text in the revised paper is formatted in italic bold,** *text from the original manuscript in italic.* We will refer to line numbers reported in the track-changes document.

I am not saying the model improvements are irrelevant and I agree that they should certainly be implemented into all newer versions of Delft3D. However, the title promises a bit more ("improve the physics behind complex numerical models"), in the Abstract it says that "similar outcomes in other applications" can be expected, and in the Conclusions it says that "users and

developers can leverage these toolboxes to detect code-related issues" and that "the insightful analysis of the results of the optimization may offer similar outcomes in other applications".

Yet no further examples are given anywhere in the text. I am wondering if the authors can think of other areas where optimization toolboxes could help to improve lake/ocean models.

Following the Reviewer's suggestion, we provided insights into research areas where our approach might allow a better understanding of the limitations of existing parameterizations and identify opportunities for improvement. We have included the following paragraph in lines 436-444:

*"Using automated tools to identify existing flaws could also help improve the representation of other simulated processes. For instance, the impact of density stratification on vertical turbulent fluxes is imperfectly represented in turbulent schemes, and internal wave breaking is accounted for in only a few lake models, for example Simstrat by Goudsmit et al. (2002). Turbulent closures like k-ε could benefit from supervised automated calibration of parameters that are traditionally fixed based on "standard" experiments and generally poorly understood. Similarly, refining horizontal turbulent models through tuning could provide insights into whether the adopted scheme (e.g. constant coefficients or sub-grid scaling following Smagorinsky, 1963) and grid resolution are sufficient to capture the timing and location of horizontal and vertical transport processes, such as up/downwellings and gyres."*

Another example of a rather general statement that might need to be clarified: in line 28 it says:

"Therefore, some processes that are crucial in lakes, but not for the original environment, may not be correctly reproduced."

I agree that the purpose for which a model has been developed is something to keep in mind and can be important. However, in the case of penetrative shortwave radiation, some of the mentioned models already have a more advanced parameterization than Delft3D even though they have been developed for the ocean (e.g., MITgcm and ROMS; see comment above). Later in line 100, it says that "many numerical models" already employ a shortwave radiation model that includes beta. So, all in all, it sounds as if Delft3D is the outlier here. This does not imply that the results of this study are unimportant, but I think some context is missing.

We agree with the Reviewer that it is not the purpose for which Delft3D was originally developed that explains per se the parametrization of penetrative shortwave radiation currently present in the model. MITgcm and ROMS, both developed for the ocean environment, have indeed more advanced parametrization. We speculate that the original scope of the model could be one of the reasons why this flaw has never raised concerns so far. Even if Delft3D was the only model without $\beta$, the message of our work would not change. Indeed, as said above, the core of this work is to discuss a method to identify what must be improved in existing schemes.

Other minor comments:

- line 33: Briefly explain surrogate models?

We included the following lines 47-53:

*"A surrogate model, also known as emulator, is a simplified and computationally efficient empirical model (Castelletti et al., 2010) that mimics the behavior of a computationally expensive model based on real model training data. In the search of the optimal parameters, the largest computation cost is indeed related to the real model runs. Thus, fast surrogate model runs are alternated to real model runs such that a lower number of real model evaluations is needed. Surrogate models can be included in traditional optimization tools such as Bayesian calibration (Ma et al., 2024) and are highly effective in speeding up the calibration process."*

- line 43: "Many examples in THE literature"

Suggestion followed, thanks.

- line 43: "Many examples": could you name a few?

We revised the sentence as follows (lines 56-58):

*"Literature reports examples where calibration has guided modelers in regions of the parameters space that hold no physical meaning (e.g. Baar et al. 2019, well discussed by Tritthart et al. 2024)."*

- line 50:  You refer to Schwindt et al., (2022) but do not say what their study was about or why we should care about it. Give key results?

We mention Schwindt et al. (2022) twice in the introduction. We first included this reference among the examples of automated calibration of Delft3D with surrogate models, and then we reported this reference again as we believe that the authors' philosophy is similar to ours. We improved the reference to the work by  Schwindt et al. (2022) as follows:

Lines 44-47:

*"Schwindt et al. (2022) assessed the uncertainty of mixing-related model parameters through a Bayesian calibrator combined with a Gaussian process emulator"*

Lines 65-68:

*"Analyses of how these tools can foster a better understanding of the numerical results have started to appear in the literature. In the previously cited work by Schwindt et al. (2022), the authors were able to identify unrealistic model setups from the high uncertainty of the a posteriori distribution of mixing-related parameters. With a similar philosophy, here we evaluate the parameterization of the heat distribution along the water column as coded in Delft3D."*

- line 53: "has been long recognized":  some references?

We added some references (line 72).

*"Henderson-Sellers, 1986; Zolfaghari et al. 2017; Shatwell et al. 2019"*

- line 61: In large lakes, $D_s$ can also vary significantly in space (see, e.g., https://doi.org/10.1016/j.jglr.2020.03.013). Maybe this is worth mentioning? Also, "can significantly vary" is very general. Could you provide some typical values for $D_s$ throughout the season?

Thanks for the input. We improved the sentence as follows (lines 81-85):

*"In large lakes, differences between littoral and pelagic Secchi depth measurements can be of the same order of magnitude as seasonal variations, as shown, e.g., by Pothoven and Vanderploeg (2020) who found up to 4 times lower transparency along the shores of Lake Michigan compared compared to its pelagic areas. Massive algal blooms, combined with basin scale circulation, can also drive complex spatial patterns of water transparency as those observed by Rahaghi et al. (2024) in Lake Geneva, with Secchi depth ranging between < 2 and 6 > m on the same day."*

- line 111-113: It is not clear here why non-penetrative terms should usually represent a sink/be negative. You get back to this later when introducing the concept of thermal equilibrium. --> Maybe mention here that this will be explained later or give a brief explanation already?

Suggestion followed, we added *"represent a sink of heat, as will be shown later"* in line 144.

- line 125-126: Can you name other lake models where the beta term is absent so that colleagues using them become aware of these limitations? As far as I can tell, MITgcm and ROMS (two other widely-used models, although admittedly mostly in large lakes or the ocean) already include beta (see comment above).

Among the models we reviewed for this work, we found that only Delft3D has no parameterization of surface shortwave radiation absorption. These models are already listed in Section 2.1 ("Use of Secchi depth in models, lines 144-155):

*"Modified versions of the Beer law were introduced in some lake-dedicated models to account for the different absorption of heat depending on the spectral bands of solar radiation. This is the case of the General Lake Model (Hipsey et al., 2019), where the authors attribute 55% of the incident solar radiation to near-infrared (NIR) and ultraviolet (UVA, UVB) radiation heating the surface directly. A default value of 45% for $\beta$ is implemented in CE-QUAL-W2 Cole and Wells, 2015), but different values of 24 to 69% are recommended depending on the type of environment, with higher values attributed to pure waters (63%) and coastal waters (69%) and lower values (24 to 58%) to lake waters. A similar approach with different proportions (35% for NIR, 65% for PAR, UVA, UVB) was adopted by Thiery et al. 2014 to simulate Lake Kivu thermal structure with an ensemble of different lake models. Among these, models explicitly including $\beta$ are SimStrat (Goudsmit et al. 2002), LAKEoneD (Joehnk and Umlauf, 2001), LAKE (Stepanenko and Lykossov, 2005), MINLAKE96 (Fang and Stefan, 1996)."*

We added some more examples in the discussion section (lines 431-434):

*"However, we expect the best improvement to come with schemes similar to those already implemented in GOTM (General Ocean Turbulence Model, Burchard et a. 1999), ROMS (Shchepetkin and McWilliams, 2005), MITgcm (Marshall et al. 1997) or AEM3D (Dallimore, 2019), which allow different extinction coefficients for different portions of the solar radiation spectrum, following Paulson and Simpson, 1977."*

However, we discussed in our reply to the general comment, our work is not aiming at improving a specific parameterization of Delft3D nor at reviewing all lake models' heat fluxes parameterizations. The focus is on the methodology we used to detect what was wrong in the

original formulation. We believe that the references and the discussion provided are sufficient to frame the context of existing models.

- line 139: "water density" sounds more natural than "water's density"

Suggestion followed, thanks.

- lines 150-151: The wording is a bit unclear. Isn't the "overall" lake temperature, as in the total heat content, always determined by the balance between the surface fluxes unless we assume geothermal heat fluxes or other sources/sinks, e.g., river inflows? Doesn't equilibrium (surface) temperature rather mean that the net heat flux is zero?

We thank the reviewer for pointing out the potentially misleading wording. Indeed, the "overall" lake temperature dynamics also depends on its heat capacity, i.e. the heat stored in the water mass. Instead, the equilibrium temperature is given by the instantaneous "*balance between all the fluxes exchanged at the surface*", without considering heat storage, so that the net heat flux is assumed to be zero.

The point we wanted to make here (see lines 148-149 of the original manuscript) was slightly different, and simpler: at a daily scale, the net amount of heat exchanged through the surface is, on average, much smaller than the single terms of the balance. This can be seen, e.g., from the values reported in the title of subplot (a) of Figure 2, with <H_net> = 6 W/m^2 against <H_np> = 271 W/m^2.

In this context, the sentence at lines commented by the reviewer is unnecessary. Therefore, we revised the sentences at lines 183-187 as follows:

"*The total* exchanged heat $\int_{diel} \phi_{net} dt$ is typically much smaller than the two individual components on the right-hand side of equation (9), as can be seen in Figure 2a. Under these conditions, and to illustrate the behaviour of the system, it can be assumed that $\int_{diel} \phi_{net} dt \simeq 0$. Therefore, ..."

- line 175: "second term on the right-hand side of equation (13)": unclear which line of 9equation (13) you refer to. Introduce (13.1) and (13.2)?

We added *"(first line)"* to specify we refer to the first line of equation (13), where two terms are summed (line 215).

- line 187: "expressed by equation (14)": do you mean by equation (12)?

We apologise for the typo. We corrected the reference to equation 12 (line 227).

- line 212: typo in "correctly"

Thanks, we corrected the typo.

- line 213-214: It is unclear where and when exactly these temperature measurements were taken. What does "surface temperature" mean exactly? I assume measurements were taken at some depth near the surface. Could the depth of this "surface temperature" have an impact on the optimization? Maybe summarize the measurement locations and periods in a short section or table?

We thank the reviewer for raising this important point. This comment is similar to that of reviewer 1 about how sensitive surface measurements can be. We included more detail on the in-situ measurements where they are presented (Appendix B), lines 482-484:

*" In-situ water temperature profiles are sampled on a monthly basis in the middle of the lake Fig. B1b,c across irregular depth intervals: 0 (just below the surface),2,5,7,10,12,15,17,20,25,30,35,40 and 43. Monthly measurements of water transparency (expressed as Secchi depth, Fig. B1d) are also recorded in the same observation point."*

We added a paragraph in the discussion section dealing with this aspect, *lines 422-429:*

*" When evaluating the model performance based on water temperature just below the surface, it is worth recalling that the in situ observation of such temperature is sensitive to many factors: from the plunging depth of the instrument, to water level fluctuations modifying the actual water depth of e.g. permanent thermistor chains, to the exposure of the near-surface sensor to direct sunlight (Bärenbold et al., 2024). In this regard, only detailed vertical resolution of in situ measurements can support understanding of how well the surface temperature microstructure is reproduced by the model."*

- line 219: You optimize based on full-depth or surface temperatures. As shortwave radiation penetration decays exponentially, I assume the upper few cm or m are most strongly impacted by the improved parameterization. Would it make sense to define a cost function that focuses on these upper layers? For example, by taking one e-folding length scale as the range of the cost function. Or do you expect the results to be insensitive to this?

We thank the reviewer for this comment. It is absolutely true that the upper layers are the most impacted by the improved parameterization. As it is shown in Fig. 4a-b, the temperature profile simulated with the original (a) and the modified (a) scheme does not change significantly below 2 m depth. Similarly, Fig. 3 shows that the largest impact of including $\beta$ is on the objective function $\epsilon_{T0}$ (panel c), which measures the error at the surface. If finer vertical resolution is available from in-situ measurements, we could define a cost function considering the upper layers until e.g. the Secchi depth, and emphasize the relative importance of the near-surface layers with an exponential weight function. We did not follow this approach in this work as data is only available at 0 and 2m in Lake Morat. We added a paragraph in the discussion section dealing with this aspect (lines 429-431):

*"If a good level of detail is available from measurements, more sophisticated objective functions could also be used to emphasize the relative importance of the near-surface layers with e.g. exponentially decaying weights associated with the extinction coefficient".*

- line 241: If you give the computation time, also give the number of nodes/cores.

We included the required additional information in lines 292-293:

*"The simulation time for 5 months was about 11 hours on a High Performance Computing Cluster. Each simulation utilized 6 cores on a single node, resulting in 72 CPU cores being used per iteration."*

- lines 255-259: Can you comment on when/why it is valid to neglect advection and diffusion and the role of the wind speed? I assume low wind means less wind-induced turbulent mixing and advection, so vertical temperature gradients due to wrong surface forcing can build up more easily, driving stronger "artificial" convection in the model to compensate for that.

In a real lake environment we never recommend to neglect advection and diffusion to correctly reproduce surface temperature dynamics. It is certainly true that in low wind conditions the impact of the inaccurate parameterization in the original Delft3D model version is more evident than during windy days. We addressed this aspect in lines 404-410 and we thank the reviewer for raising this point:

*"We have shown in Figure 4 that the largest difference between the two model configurations is at the surface, and in particular during daylight hours in the stratified period. In the temperature profiles shown, a strong gradient is present between the near-surface temperature (i.e. 0 m) and the layer immediately below (2 m). On other days, when the measured surface temperature is relatively well mixed with the layers below, the difference between the two model versions is not that relevant. In fact, if enough mixing is physically provided by, for example, strong wind-induced turbulence, the model does not need to generate artificial convection to compensate for the negative buoyancy at the surface caused by the incorrect parametrization."*

- line 276: "fully consistent with literature values": Could you please also give these literature values here and not only in the caption of Figure 3?

We actually reported literature values in Section 2.1 ("Use of Secchi depth in models, lines 144-155, see also our reply to the comment to lines 125-126 from original manuscript). We improved the link with the literature already presented by summarizing the range limits and referring to that section in line 333 as follows:

*"fully consistent with the range of literature values (20-60%) presented in Section 2.1."*

- line 287: "ON three selected days"

Suggestion followed.

- line 287: Why did you select these days, and does that choice matter for the results? For example, do you expect different results for calm vs. windy conditions (see comment above on wind-induced advection and turbulent diffusion)?

We presented the results in days when a strong surface stratification is visible from the measured temperature profiles. In days when the measured surface temperature was relatively well mixed with the layers below the difference between the two model versions is not that relevant. We addressed this aspect in lines 404-410 (see above) and we thank the reviewer for raising this point.

- lines 329-336 (see also lines 290-291): Does this imply that the level of improvement one can expect from the improved representation of shortwave penetration directly depends on the vertical grid spacing near the surface? If so, this seems worth mentioning/discussing further.

We thank the reviewer for pointing this out. The finer the grid spacing, the more detailed the microstructure can be represented, the more impactful will be the parameterization adopted for

capturing the penetration of shortwave radiation. However, as we already replied to the comment to lines 219, finer resolution in the observed data would be also needed. It is likely that if we used a thicker surface layer (e.g. 60 cm thick) the value of the $\delta$ parameter would have been closer to 1 than the 0.5 value we obtain. We addressed this aspect together with the other points related to vertical resolution and shape of the objective function in lines 422-435:

*"It is therefore likely that if we used a thicker surface layer (e.g. 60 cm thick), the value of the parameter would be closer to 1.*

*The choice of the thickness of the surface layer is a crucial task, which depends not only on the level of accuracy we desire from the model but also on how much we trust the in situ measurements. When evaluating the model performance based on water temperature just below the surface, it is worth recalling that the in situ observation of such temperature is sensitive to many factors: from the plunging depth of the instrument, to water level fluctuations modifying the actual water depth of e.g. permanent thermistor chains, to the exposure of the near-surface sensor to direct sunlight (Barenbold et al. 2924). In this regard, only detailed vertical resolution of in situ measurements can support understanding of how well the surface temperature microstructure is reproduced by the model. If a good level of detail is available from measurements, more sophisticated objective functions could also be used to emphasize the relative importance of the near-surface layers with e.g. exponentially decaying weights associated with the extinction coefficient. However, we expect the best improvement to come with schemes similar to those already implemented in GOTM (General Ocean Turbulence Model, Burchard et a. 1999), ROMS (Shchepetkin and McWilliams, 2005), MITgcm (Marshall et al. 1997) or AEM3D (Dallimore, 2019), which allow different extinction coefficients for different portions of the solar radiation spectrum, following Paulson and Simpson, 1977. This further improvement is beyond the scope of our work and we encourage developers to enhance the Delft3D source code to optimize its performance on the surface of water bodies."*

General minor comments:

- Not sure if this is on purpose or just a formatting error: When giving the units, there is a full stop/period instead of a space between units, e.g., W.m-2, instead of W m-2

We thank the reviewer for noting this formatting error, which we removed.

- Write "in Appendix A/B", instead of just "in A/B"?

Suggestion followed

- Consistent AE, BE spelling (modelling and modeling are both used)

We are sorry for the inconsistency and we revised the text and assured consistency with AE spelling throughout the text.

- There are a few minor English mistakes, as pointed out by the other reviewer. I trust that the authors will correct them in the revised version

We are sorry for the mistakes and we revised the text thoroughly.